# Formation and Maturation of the Phagosome: A Key Mechanism in Innate Immunity against Intracellular Bacterial Infection

**DOI:** 10.3390/microorganisms8091298

**Published:** 2020-08-25

**Authors:** Hyo-Ji Lee, Yunseo Woo, Tae-Wook Hahn, Young Mee Jung, Yu-Jin Jung

**Affiliations:** 1Department of Biological Sciences and Kangwon Radiation Convergence Research Support Center, Kangwon National University, Chuncheon 24341, Korea; koko7912@kangwon.ac.kr (H.-J.L.); yunseo@kangwon.ac.kr (Y.W.); 2College of Veterinary Medicine, Kangwon National University, Chuncheon 24341, Korea; twhahn@kangwon.ac.kr; 3Department of Chemistry and Kangwon Radiation Convergence Research Support Center, Kangwon National University; Chuncheon 24341, Korea; ymjung@kangwon.ac.kr

**Keywords:** phagocytosis, phagosome maturation, pathogen, lysosome, bacterial infection, innate immunity, innate defense mechanism

## Abstract

Phagocytosis is an essential mechanism in innate immune defense, and in maintaining homeostasis to eliminate apoptotic cells or microbes, such as *Mycobacterium tuberculosis*, *Salmonella enterica*, *Streptococcus pyogenes* and *Legionella pneumophila*. After internalizing microbial pathogens via phagocytosis, phagosomes undergo a series of ‘maturation’ steps, to form an increasingly acidified compartment and subsequently fuse with the lysosome to develop into phagolysosomes and effectively eliminate the invading pathogens. Through this mechanism, phagocytes, including macrophages, neutrophils and dendritic cells, are involved in the processing of microbial pathogens and antigen presentation to T cells to initiate adaptive immune responses. Therefore, phagocytosis plays a role in the bridge between innate and adaptive immunity. However, intracellular bacteria have evolved diverse strategies to survive and replicate within hosts. In this review, we describe the sequential stages in the phagocytosis process. We also discuss the immune evasion strategies used by pathogens to regulate phagosome maturation during intracellular bacterial infection, and indicate that these might be used for the development of potential therapeutic strategies for infectious diseases.

## 1. Introduction

Phagocytosis is a mechanism for the uptake and digestion of large particles (0.5 > μm) in vacuoles surrounded by the plasma membrane [1,2]. This evolutionarily conserved process is used by unicellular organisms, such as *Dictyostelium discoideum*, to obtain food and nutrients [3]. In multicellular organisms, phagocytosis plays a role in maintaining cellular homeostasis by removing dead cells, microbes and cellular or foreign debris [4,5]. It is also a fundamental defense mechanism that activates immune and inflammatory responses against invading pathogens. In mammals, phagocytosis is mediated by professional antigen-presenting cells (APCs), including macrophages, monocytes, dendritic cells and neutrophils. However, nonprofessional phagocytic cells, such as epithelial cells [6], endothelial cells [7], fibroblasts [8] and B cells [9], are also involved in phagocytosis. These cells recognize foreign pathogens and internalize them into phagosomes.

Professional phagocytes express a wide range of phagocytic receptors at the membrane surface, and these can recognize pathogen-associated molecular patterns (PAMPs) presented by foreign pathogens or target particles [10]. After engagement between phagocytic receptors and ligands, phagocytes induce intracellular signaling cascades and activate a rearrangement of the actin cytoskeleton, and during phagocytosis, this rearrangement of the actin cytoskeleton triggers the formation of pseudopods that surround the target particles and seal the distal tip of the phagocytic cup, which results in the internalization of target particles into a nascent phagosome [11,12].

The newly formed phagosome undergoes a series of fusion and fission events with endocytic organelles through a ‘kiss-and-run’ mechanism, to acquire the molecules necessary for each phagosome stage [13]. Phagosomes then interact with lysosomes to form phagolysosomes, resulting in the degradation of the phagosomal contents. These processes are called ‘phagosome maturation’ [14].

Phagocytosis is an essential process for the elimination of dying cells or microbial pathogens. However, this process is tightly regulated and requires complex crosstalk between intracellular compartments. This review discusses the overall process of phagocytosis as an innate immune defense mechanism for the removal of microbial pathogens by professional phagocytes. We also focused on a novel molecular finding regarding the regulation of phagosome maturation during microbial pathogen infection.

## 2. Initiation of Phagocytosis

### 2.1. Recognition of Foreign Particles

Phagocytes need to bind to microbial pathogens for the induction of phagocytosis during microbial pathogen infection. Phagocytes express a variety of phagocytic receptors on the membrane surface, and these include pattern-recognition receptors (PRRs), opsonic receptors and receptors for apoptotic cells (Table 1). Several receptors are co-expressed in a single phagocyte and cooperate to recognize and digest microbial pathogens. Not all receptors involved in phagocytosis are phagocytic receptors. Several receptors, such as Toll-like receptors (TLRs) and some G-protein coupled receptors, play a role in the priming or regulation of phagocytosis [15,16]. Although they can bind PAMPs, these receptors cannot directly engulf microbial pathogens. Microbial pathogens are recognized directly by receptors expressed on the cell surface of phagocytes that interact with PAMPs, or indirectly by receptors that interact with opsonins.

#### 2.1.1. Pattern-Recognition Receptors (PRRs)

Some receptors, including dendritic cell-associated C-type lectin-1 (dectin-1) [17], dendritic cell-specific ICAM-3-grabbing nonintegrin (DC-SIGN) [21], macrophage-inducible C-type lectin (Mincle) [24], and the macrophage C-type lectin dectin-3 (MCL) [27], can bind PAMPs and induce phagocytosis (Table 1). These receptors are members of the C-type lectin receptor (CLR) family, and are expressed on the cell surface of phagocytes. Dectin-1 binds to polysaccharides expressed in fungi, including *Candida albicans* [18], *Saccharomyces cerevisiae* [19] and Pneumocystis carinii [20]. Dectin-1 has an immunoreceptor tyrosine-based activation motif (ITAM)-like motif in the cytoplasmic tail [44], and this motif needs to be phosphorylated to activate intracellular signaling, and thus induce various cellular responses, including phagocytosis. The transfection of dectin-1 induces phagocytic uptake in 3T3 fibroblasts, which are nonprofessional phagocytic cells, and this finding indicates that dectin-1 can serve as a phagocytic receptor during phagocytosis [19]. Dectin-1 also associates with another phagocytic receptor, complement receptor 3 (CR3), for phagocytosis [45]. DC-SIGN can recognize mannosyl glycans and fucosylated glycans derived from viruses, bacteria, and fungi [22]. It was initially believed that the expression of DC-SIGN is restricted to only dendritic cells [46], but recent findings suggest that DC-SIGN is expressed in selected macrophage populations [47], and in the endothelium [48]. Serrano-Gomez et al. showed that DC-SIGN mediates the recognition and internalization of *Aspergillus fumigatus* conidia in dendritic cells and macrophages [49]. It has also been demonstrated that DC-SIGN binds to mannose-capped lipoarabinomannan (ManLAM) derived from *Mycobacterium tuberculosis*, and mediates the internalization of *M. tuberculosis* in human dendritic cells and monocyte-derived macrophages [23], which indicates that DC-SIGN plays a role as the major phagocytic receptor for *M. tuberculosis*. Pei Zhang et al. showed that HeLa-DC-SIGN cells, which are HeLa cells stably transfected with DC-SIGN, interact with and phagocytose *Escherichia coli* [50], and this finding suggests that DC-SIGN participates in the recognition of *E. coli*, which results in phagocytosis. Mincle was first described as a receptor for trehalose-6,6′-dimycolate (TDM), derived from mycobacterial cell wall glycolipids [25], and for Sin3-associated protein 130 (SAP130) released from damaged and necrotic cells [26]. Atul Sharma et al. found that the phagocytosis of GFP-labeled *Klebsiella pneumoniae* was significantly reduced in *mincle*-knockout (*mincle*^−/−^) neutrophils [51], which indicates that Mincle plays an important role in bacterial recognition and uptake by neutrophils. MCL also interacts with mycobacterial TDM, as well as fungal β-glucans [28]. Lisa M. Graham et al. showed that MCL induces phagocytosis via spleen tyrosine kinase (Syk) in myeloid cells, although MCL lacks a signaling motif in the cytoplasmic tail [52]. MCL mediates TDM-induced Mincle expression [53] and cooperates with Mincle for the induction of phagocytosis [54].

Other PRRs, including CD14, mannose receptor, scavenger receptor A (SR-A), CD36 and MARCO, also contribute to phagocytosis, but whether these receptors can directly trigger phagocytosis or contribute to priming, or regulate phagocytosis together with other phagocytic receptors, remains unclear [55]. CD14 is a receptor for LPS-binding protein (LBP) [29], and is responsible for the uptake of *M. tuberculosis* [56], *Cryptococcus neoformans* [57] and *E. coli* [58]. Ivan Zanoni et al. demonstrated that CD14 regulates the LPS-induced endocytosis of TLR4 [59]. The mannose receptor binds to mannan, and recent studies have shown that the mannose receptor mediates the bacterial uptake of pathogenic and nonpathogenic mycobacteria [30], and regulates the mycobacteria-induced phagocytosis of apoptotic cells [31]. SR-A binds to LPS derived from gram-negative bacteria or some gram-negative bacteria [32], and CD36 interacts with *Plasmodium falciparum*-infected erythrocytes [33]. Macrophage receptors with collagenous structures (MARCOs) recognize various bacteria and oxidized lipids in human macrophages [34,35], and Tennille Thelen et al. found that the phagocytosis of *Clostridium sordellii* is decreased in *marco*^−/−^ mice [36].

#### 2.1.2. Opsonic Receptors

Opsonization is the mechanism for the labeling of foreign particles, microbes and apoptotic cells with opsonins, such as immunoglobulins (Igs) or complement components. Phagocytes can recognize target particles labeled with opsonins via their receptors on the cell surface. In this context, opsonins play a role as bridges between phagocytes and foreign particles. Fcγ receptors (FcγRs) are expressed on leukocytes, and bind to the Fc portion of IgG. There are three broadly characterized types of FcγR with different binding affinities: FcγRI (CD64), FcγRII (CD32) and FcγRIII (CD16) [37]. FcγR contains an ITAM in the cytoplasmic domain, which is phosphorylated by Src family tyrosine kinases (SFKs), to induce and activate signal transduction during FcγR-mediated phagocytosis. These receptors interact with the Fc portion of Ig in Ig-coated foreign particles, and phagocytes then internalize and digest the IgG immune complex; this process is called antibody-dependent cellular phagocytosis (ADCP) [60]. As an example, Atsuhiro Masuda et al. demonstrated that the phagocytosis of bacteria opsonized with IgG is significantly diminished in macrophages without FcγR during *Citrobacter rodentium* infection [61], indicating that FcγR plays a critical role in the control of bacterial infection by inducing antibody-mediated phagocytosis. Complement receptors (CRs) are G-protein coupled receptors (GPCRs) expressed on several immune cells, and recognize components of complements, including C1q, C4b, C3b and iC3b. Foreign particles induce the generation of C3b and other bound cleavage products that bind to various complement receptors, such as CR1, CR2, CR3, CR4 and CRIg [62]. The interaction between CRs and complement-opsonized foreign particles leads to receptor clustering and phagocytosis [63]. In particular, CR3 recognizes a variety of ligands, including iC3b on complement-opsonized target particles, ICAM-1, ICAM-2, fibronectin, LPS, oligodeoxynucleotides and zymosan [38]. CR3-mediated phagocytosis enhances bacterial clearance, such as that of *Pseudomonas aeruginosa*, *Salmonella* and *E. coli* [64].

#### 2.1.3. Receptors for Apoptotic Cells

Phagocytes distinguish and selectively eliminate dying cells among healthy living, apoptotic, and necrotic cells. The clearance of apoptotic cells is essential for the maintenance of homeostasis, for the development of normal organs, and for the removal of foreign pathogens [5,65]. During apoptosis, dying cells release ‘find me’ signals, such as sphingosine 1-phosphate (S1P), thrombospondin (TSP), CX3CL1, lysophosphatidylcholine (LPC), ATP and UTP, and these molecules induce the recruitment of phagocytes and interact with different types of specific receptors. To ensure their specific identification, dying cells also expose ‘eat me’ signals on the cell membrane [66]. Specifically, dying cells enhance the exposure of phosphatidylserine (PS), which is an inner membrane lipid, for recognition by specific receptors on phagocytes. In addition, dying cells also trigger the conversion of the surface charge of glycoproteins and lipids on the plasma membrane, the expression of ICAM-3 and oxidized low-density lipoprotein (oxLDL)-like molecules, and the opsonization of apoptotic cells by the complement system [5]. Subsequently, phagocytes physically internalize dying cells through receptor-mediated signaling and the rearrangement of the cytoskeleton. PS binds to various receptors, including integrin αVβ3, TIM-1, TIM-4, brain-specific angiogenesis inhibitor 1 (BAI1), and stabilin-2 [39]. oxLDL-like molecules interact with CD36, a scavenger receptor [40]. Michael E. Greenberg et al. demonstrated that the accumulation of apoptotic cells is significantly increased in *cd36*^−/−^ mice [41], which indicates that CD36 interacts with apoptotic cells to remove these cells via phagocytosis. Devitt et al. found that the neutralization of CD14 using the anti-CD14 monoclonal antibody (mAb) 61D3 inhibits the clearance of PS-containing liposomes [42]. CD14 can recognize PS on apoptotic cells, and CD91 recognizes the complement protein C1q, which directly binds to apoptotic cells [43]. Intracellular signaling cascades through these receptors, to initiate and activate the phagocytosis of dying cells.

### 2.2. Phagocytic Receptor Signaling for the Formation of Nascent Phagosomes

Engagement between phagocytic receptors and ligands derived from invading pathogens or apoptotic cells triggers the initiation and activation of various intracellular signaling pathways to internalize foreign particles through phagocytosis. These phagocytic signals induce the rearrangement of the actin cytoskeleton and cause the depression of the membrane to form a phagocytic cup. Subsequently, pseudopods, which are transient membrane protrusions that extend around a foreign particle, are formed, and completely enclose the foreign particles to develop a nascent phagosome. The most well-known receptors for the intracellular signaling of phagocytosis are Fc receptors and CRs.

#### 2.2.1. Fc Receptor (FcR) Signaling

Following engagement between FcγR and IgG, tyrosine residues within ITAM on the cytoplasmic domain of FcγR are phosphorylated by Src family tyrosine kinases, including Lyn, Lck and Hck (Figure 1). These phosphorylated residues are utilized as docking sites for Syk, that induce the phosphorylation of diverse substrates and the initiation of multiple intracellular signaling pathways for phagocytosis [67]. Cheryl J. Fitzer-Attas et al. demonstrated that IgG-dependent phagocytosis is significantly reduced in *hck*^−/−^
*fgr*^−/−^ and *lyn*^−/−^ bone marrow-derived macrophages (BMDMs), which suggests that FcγR-mediated phagocytosis requires the participation of various SFKs [68].

Because Syk is connected to a docking site of ITAM, it leads to the activation of many signaling molecules, including linker for activation of T cells (LAT), protein kinase C (PKC), phospholipase A2 (PLA2), phospholipase Cγ (PLCγ), phospholipase D (PLD), phosphatidylinositol 3 kinase (PI3K), extracellular signal-regulated kinase (ERK), and the small GTPases of the Rho family (Rac1/2 and Cdc42) [69]. LAT acts as a scaffold protein that recruits and interacts with other proteins by interacting with activated Syk [70]. Activated Syk activates Vav, a guanine nucleotide exchange factor (GEF), that acts as an exchange factor for Rac1 and Cdc42. Both Rac1 and Cdc42 regulate the actin nucleation complex Arp2/3 involved in actin polymerization [71]. The activation of Rac1 and Cdc42 regulates the activation of the NF-κB and JNK pathways and rearrangement of the actin cytoskeleton, and these effects trigger the formation of pseudopods to internalize foreign particles. Syk also activates PI3K, which phosphorylates PI(4,5)P_2_ to generate PI(3,4,5)P_3_ at the phagocytic cup. Phosphorylated Syk recruits PLCγ, which in turn produces diacylglycerol (DAG) and inositol triphosphate (IP_3_) from PI(4,5)P_2_. DAG induces the activation of PKC, which activates the ERK1/2 and p38 MAPK signaling pathways. IP_3_ enhances the intracellular release of Ca^2+^ from the endoplasmic reticulum (ER) to the cytoplasm surrounding the phagocytic cup. Intracellular Ca^2+^ also regulates the rearrangement of the actin cytoskeleton [72], and activates the nuclear factor of activated T cells (NFAT), which induces the expression of various cytokines involved in inflammation and immune regulation, including tumor necrosis factor α (TNF-α), interleukin-6 (IL-6), IL-2, IL-10, IL-8 and interferon-γ (IFN-γ) [73].

#### 2.2.2. Complement Receptor (CR) Signaling

Complements are circulating proteins in the blood and body fluids. Complements are not constitutively active, but are sequentially activated by an enzyme cascade. Specifically, these proteins are activated by proteolytic cleavage of the subsequent protein, through the activation of one protein. Upon infection with foreign pathogens, complements bind to the surface of pathogens and induce complement cascades. Complement components bound to pathogens are recognized by CRs expressed on phagocytes [74]. Among CRs, CR3 (integrin αMβ2) has excellent phagocytosis efficacy. CR3 interacts with the major soluble complement component C3b and the inactive derivative iC3b [63]. Unlike FcR-mediated phagocytosis to form pseudopods, CR-mediated phagocytosis is a ‘sinking’ phagocytosis process that internalizes foreign pathogens through membrane depression [10]. The actin and microtubule cytoskeletons are used to engulf foreign pathogens in ‘sinking’-mediated manners during CR-mediated phagocytosis [15]. Following binding of the complement component and CR, the GTPase Rho, particularly RhoA, is activated via various intracellular signaling pathways, and activated Rho leads to F actin polymerization (Figure 2) [75]. In turn, Rho activates Rho-associated kinase (ROCK), the Arp2/3 complex and mammalian diaphanous-related formin 1 (mDia1). Activated ROCK phosphorylates myosin II, resulting in the accumulation of Arp2/3 and actin assembly. Activated Rho also triggers the accumulation of mDia1, which also contributes to actin assembly in phagocytic cups. mDia1 binds to the microtubule-associated protein cytoplasmic linker protein of 170 kDa (CLIP-170), which coordinates actin polymerization [76].

In the past, CR-mediated phagocytosis was thought to be activated by mechanisms distinct from those of FcR-mediated phagocytosis, because F Kiefer et al. showed that CR-mediated phagocytosis is normally induced in *syk*^−/−^ mice [77]. In addition to this study, E Caron et al. found that Rho GTPases are recruited to phagosomes surrounding complement-opsonized particles [78], and that only Rho, but not Rac and Cdc42, colocalizes with F actin in CR3-expressing COX cells. These results suggest that CR-mediated phagocytosis needs the GTPase Rho for the rearrangement of F actin. However, recent studies have shown that similar to FcR-mediated phagocytosis, CR-mediated phagocytosis activates intracellular signaling pathways. Using C3bi-opsonized zymosan, Yuhong Shi et al. demonstrated that Syk, a downstream molecule of FcR-mediated phagocytosis, is phosphorylated during CR3-mediated phagocytosis induced by complement activation [79]. These researchers also found that the knockdown of Syk with siRNA inhibits CR3-mediated phagocytosis, which suggests that Syk is required for CR-mediated phagocytosis.

## 3. Phagosome Maturation

Following the formation of nascent phagosomes, phagosomes sequentially acquire the necessary proteins for maturation, through a series of fusion and fission events with endocytic organelles (early or sorting endosomes, late endosomes, and lysosomes), because early phagosomes cannot remove foreign particles. This process is known as ‘phagosome maturation’ (Figure 3). The stages of phagosome maturation include the early, intermediate, and late phagosome and the phagolysosome. During phagosome maturation, early phagosomes are gradually acidified to generate a degradative environment for the destruction of foreign particles and the delivery of a variety of proteins, including Ras-associated binding GTPase (Rab GTPase), vacuolar ATPase (V-ATPase), acid hydrolases, acidic proteases and major histocompatibility complex (MHC) class II molecules. Finally, phagosomes fuse with lysosomes to produce phagolysosomes, which have acidic and oxidizing environments, and can degrade their contents using a variety of hydrolytic enzymes. Phagosome maturation is required for the coordinated action of Rab proteins, including Rab5 and Rab7 [80]. These proteins act as central regulators to control various steps in endosomal and phagosomal processes. The following section describes the detailed events underlying the molecular mechanism of phagosome maturation.

### 3.1. Early Phagosome

Phagosome maturation is initiated after the newly formed phagosome is separated from the cell membrane. The initial step in the formation of early phagosomes is the interaction with early endosomes. Early endosomes, also known as sorting endosomes, serve as sorting stations characterized by mild acidic environments (pH ~6.1) and deficient hydrolytic activity [81]. Early endosomes express specific proteins, such as Rab5, early endosomal antigen 1 (EEA1), soluble *N*-ethylmaleimide-sensitive factor-attachment protein receptor (SNARE) and PI(3)P kinase. After interaction with the early endosome, the early phagosome acquires GTP-bound Rab5, an active form, EEA1, and a small number of proton pumping v-ATPases (Figure 4) [1]. The early phagosome has a near-neutral pH of approximately 6.3, due to the low accumulation of v-ATPase. Recent studies suggest that the early endosome has Rab22a and GTPase exchange factors (GEFs), such as Rabex-5 and Rabaptin-5 [82]. Rab22a triggers the recruitment of the Rab5 GEF Rabex-5 to promote the activation of Rab5 on early phagosomes. The GEF Rabex-5 plays a role in the exchange of GDP for GTP, which results in the connection of GTP-bound Rab5 and the recruitment of other proteins in early phagosomes [83]. Rabaptin-5 recruits class III phosphoinositide 3-kinase vacuolar protein-sorting 34 (Vps34) and enhances Rabex-5 activity, which results in the formation of a positive feedback loop for Rab5 activation [84]. Vps34 induces the generation of PI(3)P, which plays an important role in diverse events during early phagosome progression. PI(3)P and GTP-bound Rab5 recruit EEA1 [85]. EEA1 promotes fusion with the early phagosome and the early endosome, by tethering the target membrane. Rutilio A. Fratti et al. demonstrated that the blockade of Vps34 or EEA1 with a specific antibody or pharmacologic agent suppresses the maturation of a phagosome containing latex beads [86]. EEA1 also contributes to fusion by interacting with the SNAREs syntaxin 6 and syntaxin 13. PI(3)P also recruits the class C core vacuole/endosome tether (CORVET) complex, which enhances fusion by tethering membranes [87]. The CORVET complex is another Rab5 effector containing four core subunits (Vps11, Vps16, Vps18, and Vps33) and two specific subunits (Vps3 and Vps8). In particular, the CORVET core subunit Vps33 facilitates the recruitment of soluble SNAREs in the phagosomal membrane, to enhance membrane fusion [88].

During the maturation of early phagosomes, several components are recycled to the plasma membrane, or delivered to the *trans*-Golgi network (TGN), via Rab4, Rab11, Rab10 and the retromer complex [81]. Rab11 and Rab4 mediate recycling trafficking to the plasma membrane. Rab11 also contributes to recycling traffic to the TGN. The retromer complex consists of sorting nexin (SNX) dimers (SNX1 or SNX2 and SNX5 or SNX6) and cargo recognition Vps components (Vps26, Vps29 and Vps35), which deliver recycled phagosomal components to the TGN [89].

Recently, Jennifer Martinez et al. demonstrated that microtubule-associated protein 1 light chain 3 (LC3), which forms part of the autophagy machinery, is recruited to the phagosome, surrounded by a single membrane to eliminate dead cells [90]. This process is termed LC3-associated phagocytosis (LAP). Similar to phagocytosis, LAP is also initiated through the recognition of specific phagocytic receptors, such as TLR, FcγRs, TIM-4 and dectin-1 [91]. During receptor-mediated phagocytosis, PI(3)P is generated by a class II PI3K complex consisting of VPS34, VPS15, Beclin-1, ultraviolet radiation resistance-associated gene protein (UVRAG) and Rubicon. PI(3)P binds to the p40^phox^ subunit of the phagocytic NADPH oxidase (NOX) complex, and then induces the recruitment of gp91^phox^, p22^phox^, p47^phox^, p67^phox^ and Rac. The production of ROS recruits autophagy-related genes (ATGs), including Atg7-Atg3, Atg5-Atg12 and Atg16L1, which link LC3-I to phosphatidylethanolamine (PE) on the phagosomal membrane, to form LC3-II [92]. Finally, LC3-decorated phagosomes fuse with lysosomes to remove internalized pathogenic bacteria, fungi, parasites and dying cells [93]. However, why LC3 accumulates in the phagosome and mediates the induction of LAP remains unclear. In addition, the conditions and frequencies that trigger LAP are poorly understood. It has only been reported that the frequency of LC3-decorated phagosomes is approximately 40-80% of total phagosomes in mice and only 5-10% of total phagosomes in human cells. LAP induces different outcomes, including the promotion of phagosome maturation, the modulation of inflammation, the delivery of internalized molecules to PRR-containing vesicles, and the enhancement of antigen presentation. Therefore, LAP, which is regulated depending on the cellular context, is emerging as a new therapeutic target.

### 3.2. Late Phagosome

As phagosome maturation progresses, the PI(3)P-enriched early phagosome interacts with PI(4)P-enriched late endosomes, to develop the late phagosome, which is a more acidic compartment. PI(4)P, synthesized by phosphatidylinositol 4-kinase 2a (PI4K2A), is needed for the recruitment or activation of Rab7 and acts as a substrate for the generation of PI(4,5)P_2_ on phagolysosomes [94]. To mature into late phagosomes, the early phagosome induces the transition of Rab5 to Rab7 on the phagosomal membrane. Both Rab5 and Rab7 contribute to the transition from early to late phagosomes. Rab7 is a marker for late phagosomes and facilitates phagosome fusion with late endosomes, as well as lysosomes [95]. Similar to Rab5, Rab7 is also activated by GEFs, such as Mon1 and Ccz1, and exchanged from Rab5 to GTP-bound Rab7 on phagosomal membranes (Figure 5). Although the transition from Rab5 to Rab7 is important for phagosome maturation, the mechanism inducing and mediating this transition is poorly understood. However, recent studies suggest that the mechanisms underlying the conversion of Rab5 to Rab7 are mediated by two closely regulated steps. First, Mon1 replaces Rabex-5 in the phagosomal membrane and terminates the positive feedback loop of Rab5 activation (Figure 5A) [96]. Mon1 binds to Ccz1, and this dimeric complex acts as a Rab7 GEF. The Mon1-Ccz1 complex sequentially interacts with GTP-bound Rab5, which separates Rab7 from guanosine diphosphate (GDP) dissociation inhibitor (GDI), and allows the recruitment of a number of Rab7 molecules to the phagosomal membrane. Jason M. Kinchen and Kodi S. Ravichandra demonstrated that the Mon1-Ccz1 complex enhances the recruitment of Rab7, which suggests that the Mon1-Ccz1 complex plays a role as a bridge that mediates the transition of Rab5 to Rab7 on phagosomal membranes [97]. Second, the transition from Rab5 to Rab7 is also mediated by homotypic fusion and vacuole sorting (HOPS), which consists of a six-subunit tethering complex, activating Rab7 (Figure 5B) [14]. HOPS shares four core subunits (Vps11, Vps16, Vps18, and Vps33) with CORVET. During the formation of late phagosomes, Vps3 and Vps8 of CORVET are exchanged with Vps39 and Vps41 of HOPS, which allows the binding of Rab7 to phagosomal membranes. Accordingly, HOPS facilitates the transition from Rab5 to Rab7 by tethering phagosomal maturation [87]. Additionally, SNARE proteins, including vesicle-associated membrane protein 7 (VAMP7), VAMP8, syntaxin 7, and syntaxin 8, bring late phagosomes and lysosomes into close proximity, to promote the fusion of late phagosomes and lysosomes (Figure 5C) [98]. The late phagosome acquires lysosome-associated membrane proteins 1 and 2 (LAMP-1 and LAMP-2), which are needed for the fusion between the late phagosome and lysosome, as well as numerous hydrolytic proteins (cathepsins and hydrolases) [99]. Consequently, the late phagosome becomes more acidic (pH of approximately 5.5) and destructive, by increasing the accumulation of v-ATPase.

In addition, recycling traffic from phagosomes to the TGN, which is initiated in the early phagosome, is terminated during the maturation of the late phagosome [100]. The retromer complex is associated with GTP-bound Rab7, to terminate recycling traffic toward the TGN.

### 3.3. Phagolysosome

The final step of phagosome maturation is the formation of phagolysosomes containing large amounts of degradative components. Lysosomes are membrane-bound organelles, and between 50 and 100 lysosomes are distributed in the cytosol of mammalian cells [101]. These organelles have more than 60 acid hydrolases, which can degrade pathogens and unwanted intracellular materials. Lysosomes reach and interact with membrane-bound organelles (endocytic, phagocytic and autophagic vacuoles) though a variety of routes.

The late phagosome and phagolysosome are enriched in PI(3,5)P_2_ synthesized by the phosphorylation of PI(3)P via 1-phosphatidylinositol 3phosphate 5-kinase (PIKfyve) [102]. The role of PI(3,5)P_2_ remains poorly understood; however, this protein might modulate lysosomal calcium channels. PI(4)P on the late phagosome generates PI(4,5)P_2_, which is involved in actin polymerization and facilitates fusion between late phagosomes and lysosomes. PI(4,5)P_2_ is also converted to PI(4)P, which transfers phagolysosomes to the ER for phagosome resolution [103]. Phagolysosomes are generated through the fusion between late phagosomes and lysosomes, which requires the movement of the late phagosome and lysosome toward the microtubule-organizing center (MTOC) (Figure 6) [80].

GTP-bound Rab7 induces the recruitment of new proteins to late phagosomes, including Rab-interacting lysosomal protein (RILP) and oxysterol-binding protein-related protein 1L (ORP1L) [104,105]. RILP is a downstream adaptor of Rab7, and acts as a dynein adaptor. ORP1L regulates the binding of RILP to dynein. RILP connects Rab7-positive late phagosomes to microtubules, by inducing the recruitment of the dynein-dynactin motor complex [106]. Subsequently, Rab7-positive late phagosomes are transported toward the (−) end of microtubules, and this movement is important for the formation of phagolysosomes through the fusion between Rab7-positive late phagosomes and lysosomes [107]. Phagolysosomes have a highly acidic compartment (pH as low as 4.5), due to the transport of large amounts of protons (H^+^) to the lumen of phagosomes, via the increased accumulation of V-ATPase on phagosomal membranes. Phagolysosomes also have NADPH oxidase complexes (NOXs), which generate ROS, including superoxide anion (O_2_•^−^), hydrogen peroxide (H_2_O_2_) and hydroxyl radical (OH•), to kill foreign pathogens within phagosomes. In addition, they also contain and express a number of antimicrobial peptides with optimal activity at acidic pH, as well as hydrolytic enzymes, including various glycosidases, DNAses, cathepsins, proteases, lysozymes, and lipases [1]. Ultimately, foreign pathogens are eliminated in phagolysosomes with an environment that is suitable for the killing and degradation of these pathogens.

## 4. Regulation of Phagosome Maturation during Microbial Infection

As described above, phagocytosis maintains cellular homeostasis to eliminate invading pathogens and apoptotic cells. In particular, pathogens can survive and replicate within phagocytes, although phagocytes can express numerous antimicrobial factors, and can effectively control foreign and via phagocytosis. These pathogens include bacteria, fungi and viruses, which have various evasion strategies to avoid host defense mechanisms, including the inhibition of the PRR-mediated immune response, regulation of phagosome maturation and modulation of host cell death. In the next section, we focus on phagosome maturation among the various immune evasion strategies used by bacterial pathogens during infection.

### 4.1. Inhibition of Phagocytic Receptor-Mediated Recognition and Internalization

Phagocytes can recognize PAMPs, by using a variety of PRRs to neutralize and remove invading bacteria and can induce innate defense molecules, such as ROS, proinflammatory cytokines and inducible nitric oxide synthase (iNOS), through intracellular signaling pathways. Phagocytes can also recognize bacteria labeled with opsonins, including Igs or complements, via PRRs, scavenger receptors and FcγRs, to induce the internalization of foreign pathogens. However, some bacteria regulate the activation of innate immune defense mechanisms, using various evasion strategies to avoid their own recognition via phagocytic receptors. First, one of the strategies used by pathogenic bacteria to avoid immune recognition involves the modification of their PAMPs. Numerous studies have shown that several bacteria, including *Staphylococcus aureus* [108], *M. tuberculosis* [109], *Bacillus subtilis* [110], *Clostridium botulinum* [111] and *Streptococcus pneumoniae* [112], inhibit their own recognition via PRRs, by modifying peptidoglycans or by surrounding them with bacterial lipid components on bacterial membranes. Crystal L. Jones et al. demonstrated that *Francisella novicida* protein FTN_0757 inhibits the synthesis of bacterial lipoproteins to avoid its recognition by TLR2 [113]. It has also been reported that some bacteria, such as *Helicobacter pylori* [114], *Salmonella* Typhimurium [115], and *Yersinia pestis* [116], prevent recognition by TLR4 through the induction of the dephosphorylation of lipid A of LPS on bacterial membranes. Sara W Montminy et al. found that *Y. pestis* is not recognized by TLR4, because it induces the transformation of lipid A of LPS via acylation when grown at 37 °C (host temperature) [116]. Flagellin-expressing bacteria, including *H. pylori*, *Campylobacter jejuni* and *Bartonella bacilliformis*, prevent their recognition by TLR5, through the generation of modified flagellin, using their own virulence factors [117].

As described above, hosts can attach opsonins to foreign pathogens, which allows the effective recognition of foreign pathogens using phagocytic receptors. However, pathogenic bacteria have other evasion strategies to avoid opsonization, which prevent both recognition and phagocytosis. A number of bacteria, such as *S. aureus* [118], *S. pneumoniae* [119], *Haemophilus influenzae* [120], and *Neisseria meningitidis* [121], produce a protease to cleave Igs, which prevents the binding of Ig to bacterial membranes as opsonins. In particular, *S. pyogenes* can produce the cysteine proteinase SpeB or the endoglycosidase EndoS, which mediate the cleavage of IgG, to avoid uptake and phagocytosis by FcγR [122]. *S. pyogenes* can also bind to the Fc portion of IgG, which prevents the recognition of the Fc portion of IgG by FcγR [123]. In addition, some bacteria bind to complement factor H to inhibit the alternative complement pathway; *S. aureus*, *N. meningitidis* and *Borrelia burgdorferi* employ this strategy [124].

According to the binding of FcRs and pathogenic ligands, FcRs induce the rearrangement of the actin cytoskeleton, and the initiation of phagocytosis through intracellular signaling pathways. Several bacteria inject virulence factors with tyrosine phosphatase activity, to suppress the intracellular signaling pathway via phagocytic receptors. *S.* Typhimurium injects the tyrosine phosphatase SptP into host cells through a type III secretion system (T3SS), and this injection results in a reduction in GTPase activity [125]. In addition, SptP also regulates cytoskeletal dynamics to promote bacterial internalization into host cells. *Shigella flexneri* produces a PI(4,5)P_2_-phosphatase that diminishes phagocytosis or induces host cell death [126]. Some bacteria suppress GTPase activity by dephosphorylating GTPase-activating proteins (GAPs), or by injecting GAP mimic molecules into host cells to disrupt bacterial uptake. For instance, *Yersinia pseudotuberculosis* injects YopE, which hydrolyzes GTP-bound Rho, Rac and Cdc42 proteins [127]. Similar to *Y. pseudotuberculosis*, *P. aeruginosa* [128] and *Salmonella*
*enterica* [129] also hydrolyze GTP-bound Rho, Rac and Cdc42 proteins, to inactivate intracellular signaling pathways. In contrast, several bacteria activate the internalization pathway to facilitate their uptake into host alveolar epithelial cells [130]. *L*. *pneumophila* sequentially induces Src activation, PI3K signaling, and the activation of Rho GTPases, including Cdc42, Rac1 and RhoA, through the engagement of β1 integrin and E-cadherin. *M*. *tuberculosis* also interacts with alveolar epithelial cells to mimic the early stage of infection. The 52-kDa protein derived from *M*. *tuberculosis* plays a role in enhanced bacterial invasion and survival in HeLa cells [131].

### 4.2. Regulation of Phagosome Maturation

Phagocytosis eliminates foreign pathogens by forming phagolysosomes at the final stage of phagosome maturation. However, some bacteria can survive by using various strategies to suppress each stage of phagosome maturation within host cells. *M. tuberculosis* is the most well-known bacterium that avoids host immune responses by regulating phagosome maturation. *M. tuberculosis* has a diversity of strategies to suppress each stage of phagosome maturation: the modulation of phagosome-lysosome fusion and inhibition of phagosome acidification [132,133,134]. During *M. tuberculosis* infection, *M. tuberculosis*-containing phagosomes maintain the early phagosome state by inducing the continued accumulation of Rab5 to arrest phagosome maturation. Accordingly, *M. tuberculosis* recruits Rab22a in *M. tuberculosis*-containing phagosomes, which blocks Rab7 recruitment and arrests phagosome maturation [135]. Many studies have shown that intracellular Ca^2+^ release is required for phagosome maturation during *M. tuberculosis* infection [136,137]. However, lipoarabinomannan (LAM) derived from *M. tuberculosis* suppresses the activation of calmodulin by reducing the intracellular Ca^2+^ levels, and decreases the accumulation of PI3K Vps34 on early phagosomal membranes, to prevent the development of late phagosomes [138]. These strategies allow *M. tuberculosis*-containing phagosomes to persist as early phagosomes. Phosphatidylinositol mannoside (PIM), derived from *M*. tuberculosis, inhibits phagosomal acidification by continuously interacting with the early phagosome and the early endosome [139]. *M. tuberculosis* protein tyrosine phosphatase A (PtpA) can interact with subunit H of v-ATPase, and this interaction suppresses phagosome acidification by preventing v-ATPase tethering on the phagosomal membrane [140]. Christophe J. Queval et al. revealed that *M. tuberculosis* induces the expression of cytokine-inducible SH2-containing protein (CISH), which prevents phagosome acidification, by inducing the ubiquitination and proteasomal degradation of v-ATPase [141]. During *S*. Typhimurium infection, *S*. Typhimurium rapidly modifies phagosomes to form *Salmonella*-containing vacuoles, which allows intracellular survival. In particular, *S*. Typhimurium secretes specific effector proteins, SifA or Ssel, via the SPI2-enconded T3SS, and these proteins inhibit the fusion between phagosomes and lysosomes [142]. Similar to *M. tuberculosis*, *Neisseria gonorrhoeae* PorB, the major outer membrane porin, permits Rab5 to be continuously expressed on the early phagosome, resulting in the blockade of phagosome maturation [143]. *Coxiella burnetii* also arrests phagosome maturation by consistently expressing Rab5 on early phagosomes [144]. Marcelo G Binker et al. found that *N. gonorrhoeae* expresses an IgA1 protease that specifically cleaves LAMP-1 to arrest phagosome maturation [145]. *S. pyogenes* expresses the virulence factor M1, which suppresses the fusion between late phagosomes and lysosomes [146]. *S. pyogenes* also produces the virulence factor Mga-regulated surface protein, which blocks the accumulation of v-ATPase on phagosomal membranes to interfere with phagosome acidification [147]. Kiminori Toyooka showed that the fluorescence intensity of LysoTracker, a fluorescence dye for the detection of acidified compartments, decreases when J774A.1 cells are treated with culture supernatants of *Rhodococcus equi*, which indicates that virulence factors derived from *Rhodococcus equi* prevent phagosome acidification [148].

## 5. Conclusions

Phagocytosis maintains host homeostasis and enables the induction of rapid responses to microbial threats. Phagocytosis can occur in all types of cells, but it is most effective in professional phagocytes, including macrophages, neutrophils and dendritic cells. Phagosome formation and maturation are induced through a sequence of events, such as phagocytic receptor recognition, intracellular signal transduction, cytoskeletal rearrangement, interaction with endocytic compartments and ion transport. The final goal of these processes is to eliminate foreign pathogens and apoptotic cells. In addition, phagosome maturation affects various outcomes of innate and adaptive immune responses. Consequently, the dysfunction of phagosome maturation causes a variety of diseases, including severe infectious diseases, cancers, and Alzheimer’s disease. In this review, we describe the general aspects of each step of phagosome formation and maturation. As shown in previous studies, pathogens have evolved a range of immune evasion strategies that involve the regulation of phagosome maturation or host immune responses to survive within hosts. Furthermore, these pathogens can avoid host surveillance systems, by mimicking host molecules, or by disrupting the activation of intracellular signaling pathways. In recent years, many studies have shown the association between the phagosome maturation machinery and intracellular signaling pathways, as well as the regulation of these processes during pathogen infection. Therefore, future research could focus on the regulation of phagosome maturation by pathogens, which might help the development of novel therapeutics or therapeutic agents that eliminate pathogens and improve the host innate and adaptive immune responses during pathogen infection.

## Figures and Tables

**Figure 1 microorganisms-08-01298-f001:**
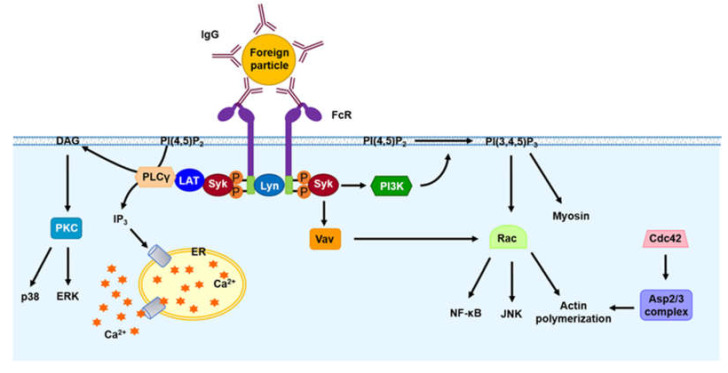
Fc receptor (FcR)-mediated signaling for phagocytosis. Engagement between FcR and the IgG immune complex induces FcR crosslinking and phosphorylation of ITAM on the cytoplasmic tail of FcR by SFKs, such as Lyn, Lck and Hck. The phosphorylation of ITAM is responsible for the recruitment and phosphorylation of Syk. Phosphorylated Syk triggers the conversion of PI(4,5)P_2_ to PI(3,4,5)P_3_ through PI3K. PI(3,4,5)P_3_ regulates the activation of Rac and myosin. Vav is activated by Syk to further activate Rac. Rac activates the NF-κB- and JNK-mediated signaling pathways. Rac and Cdc42 contribute to actin polymerization. Phosphorylated Syk also generates DAG and IP3 from PI(4,5)P_2_. DAG induces PKC-mediated activation of p38 and ERK signaling. IP_3_ triggers the release of Ca^2+^ from the ER to the cytosol.

**Figure 2 microorganisms-08-01298-f002:**
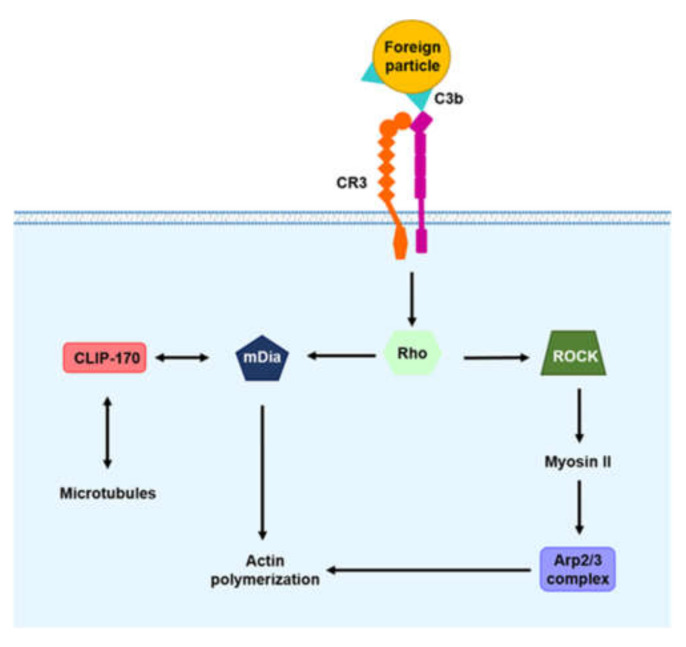
Complement receptor (CR)-mediated signaling for phagocytosis. CR binds to various complement components and initiates intracellular signaling cascades via Rho. Rho activates ROCK, and activated ROCK induces myosin II phosphorylation, and recruits the Arp2/3 complex, which leads to branched actin polymerization. Activated Rho also recruits mDia1, which mediates actin polymerization into a linear network. mDia also interacts directly with CLIP-170, which interacts with microtubules, to promote actin polymerization.

**Figure 3 microorganisms-08-01298-f003:**
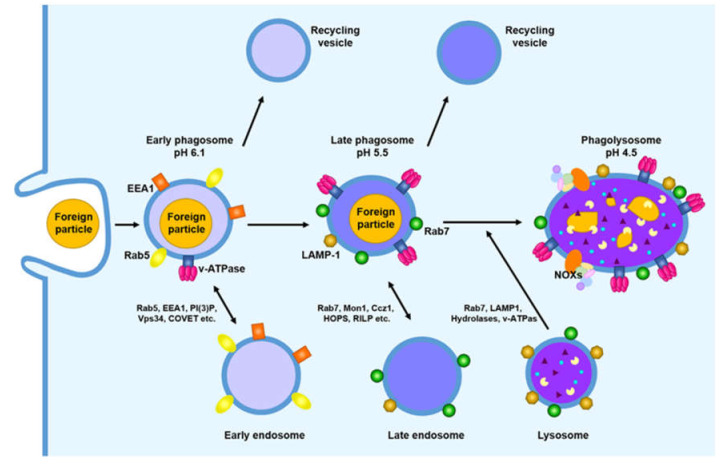
Schematic representation of phagosome maturation. Following the internalization of foreign particles, the phagosome sequentially forms early phagosomes, late phagosomes and phagolysosomes. The newly formed phagosome acquires the proteins needed for its maturation via a series of fusions and fissions with endocytic organelles (early endosome, late endosome, lysosome and recycling vesicles), and the late phagosome fuses with the lysosome for the removal of foreign particles, using various hydrolytic enzymes, antimicrobial peptides and ROS.

**Figure 4 microorganisms-08-01298-f004:**
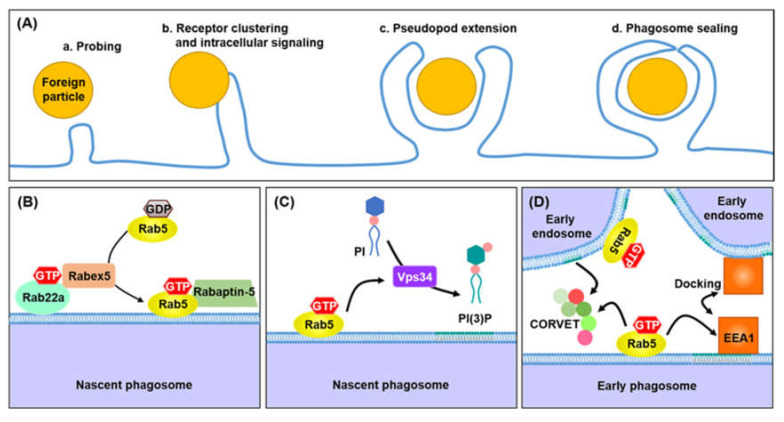
Maturation of the early phagosome. (**A**) (a) During the process of ‘probing’, the membrane expands via cytoskeletal rearrangement to recognize foreign particles using phagocytic receptors. (b) The interaction between phagocytic receptors and ligands derived from foreign particles induces receptor clusters and activates intracellular signaling pathways responsible for the formation of phagocytic cups. (c) The expansion of pseudopods is induced by the continuous assembly of actin filaments, which leads to the elongation of membranes. (d) Finally, the pseudopod surrounds the target particles and seals the distal tip of the phagocytic cup. (**B**) Following the internalization of foreign particles, the nascent phagosome begins to acquire the proteins needed for its maturation into the early phagosome. GTP-bound Rab22a recruits Rab5 GEF Rabex-5 into the phagosomal membrane. Rabex-5 converts GDP-bound Rab5 to GTP-bound Rab5, an activated form of Rab5. GTP-bound Rab5 induces the recruitment of Rabaptin-5 to further promote the activation of Rab5. (**C**) PI3K Vps34 accumulates in phagosomal membranes via Rabaptin-5, and generates phosphatidylinositol 3-phosphate (PI(3)P) from phosphatidylinositol (PI). (**D**) PI(3)P and GTP-bound Rab5 promote the recruitment of EEA1, which enhances the interaction between early endosomes and early phagosomes, by docking at the membrane of early endosomes. GTP-bound Rab5 also recruits the CORVET complex, to strengthen the fusion of early endosomes.

**Figure 5 microorganisms-08-01298-f005:**
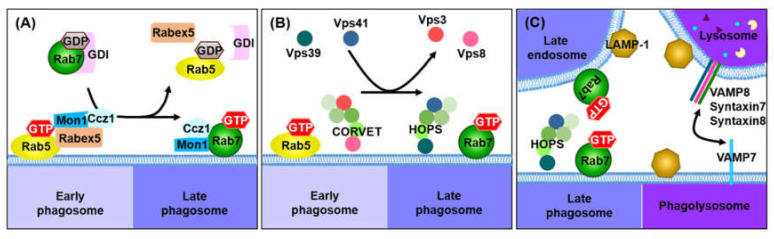
Transition from the early to the late phagosomes. (**A**) The transition of GTP-bound Rab5 to GTP-bound Rab7 is mediated by Mon1 and Ccz1. Mon1 and Ccz1 are recruited into early phagosomes to interact with Rab5. Mon1 binds to Rabex-5 to reduce Rab5 activity, and is then exchanged with Rabex-5 on early phagosomes. Mon1-Ccz1 acts as a Rab7 GEF, which induces Rab7 association and Rab5 dissociation. (**B**) The conversion of Rab5 to Rab7 is also mediated by the replacement of the CORVET complex with the HOPS complex. Vps3 and Vps8 of CORVET are exchanged by Vps39 and Vps41 of HOPS. (**C**) The HOPX complex enhances the interaction with Rab7-expressing late endosomes. Late phagosomes acquire LAMP-1 and LAMP-2 through fusion with late endosomes and lysosomes. Late phagosomes also exhibit strong interactions with late endosomes and lysosomes using SNAREs, including VAMP7, VAMP8, syntaxin7 and syntaxin8. SNARE proteins support the formation of phagolysosomes by tethering between late phagosomes and lysosomes.

**Figure 6 microorganisms-08-01298-f006:**
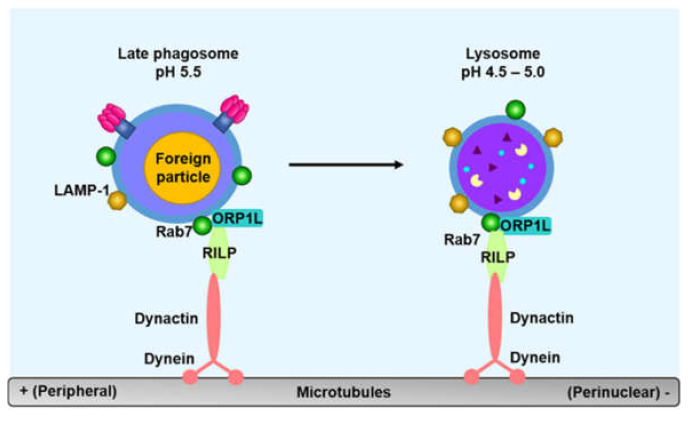
Model of the formation of phagolysosomes. For the fusion of late phagosomes and lysosomes, GTP-bound Rab7 interacts with new proteins, such as RILP and ORP1L. RILP recruits the dynein-dynactin motor complex, which moves Rab7-expressing late phagosomes toward the (−) end of the microtubule. The late phagosome then interacts and fuses with the lysosome.

**Table 1 microorganisms-08-01298-t001:** Phagocytic receptors and their ligands.

Receptors	Ligands	References
***Pattern-recognition receptors*** ***(PRRs)***		
Dectin-1	Polysaccharide derived from some yeast cells	[17,18,19,20]
DC-SIGN	Mannosyl glycans and fucosylated glycans derived from viruses, bacteria, and fungi	[21,22]
	Lipoarabinomannan	[23]
Mincle	Trehalose-6,6′-dimycolate (TDM), Sin3-associated protein 130 (SAP130)	[24,25,26]
MCL	Trehalose-6,6′-dimycolate (TDM), fungal β-glucan	[27,28]
CD14	LPS-binding protein (LBP)	[29]
Mannose receptor	Mannan	[30,31]
SR-A	LPS derived from gram-negative bacteria or some gram-negative bacteria	[32]
CD36	*Plasmodium falciparum*-infected erythrocytes	[33]
MARCO	Various bacteria and oxidized lipids	[34,35,36]

***Opsonic receptors***		
FcγRI (CD64)	IgG1, IgG3, IgG4	[37]
FcγRII (CD32a)	IgG3, IgG1, IgG	
FcγRIII (CD16)	IgG	
CR1 (CD35)	C3b, C4b, iC3b	[38]
CR2 (CD21/CD21L)	C3d, C3dg, iC3b	
CR3 (αMβ2, Mac-1, CD11b/CD18)	iC3b, C3dg, C3d, ICAM-1, ICAM-2, LPS, fibronectin, oligonucleotides, zymosan	
CR4 (CD11c/CD18)	iC3b, C3dg, C3d	
CRIg	C3b, iC3b, C3c	

***Receptors for apoptotic cells***		
Integrin αVβ3	Phosphatidylserine Phosphatidylserine Phosphatidylserine Phosphatidylserine Phosphatidylserine	[39] [39]
TIM-1		
TIM-4		
BAI1		
Stabilin-2		
CD36	OxLDL-like molecules	[40,41]
CD14	Phosphatidylserine	[42]
CD94	C1q on apoptotic cells	[43]

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
