# Peer review of "Formation and Maturation of the Phagosome: A Key Mechanism in Innate Immunity against Intracellular Bacterial Infection"

_microorganisms, 2020, doi:10.3390/microorganisms8091298_

Round 1
Reviewer 1 Report
In their review entitled “Phagosome maturation: A key mechanism in innate immunity against intracellular bacterial infection” Lee et al. provide a well-written and comprehensive summary of the literature on phagocytosis, phagosome maturation, and examples of bacterial resistance against these processes. The manuscript is well organised and scientifically sound. It provides a contribution to the field, which could be further enhanced by adding insights or perspectives to the manuscript that go beyond summarising the existing literature. One example of this is detailed below in the specific comments.
Comments:
- The manuscript describes 1) the initiation of phagocytosis, including receptors and signalling pathways involved; 2) the maturation of phagosomes, including its molecular mechanisms; and 3) examples of how bacterial pathogens prevent the initiation of phagocytosis or maturation of phagosomes. This is a fine organisation of the manuscript. However, the current title (“Phagosome maturation: A key mechanism in innate immunity against intracellular bacterial infection”) and abstract are primarily focussed on the maturation aspect. I believe a title and abstract that capture the entire scope of the review would be more in place.
- LC3-associated phagocytosis is a relatively novel process (first described by Martinez et al. in 2011, PMID 21969579) that was found to enhance the maturation process of phagosomes. I would expect this topic to be included in a review about phagosome maturation. Furthermore, it would be highly interesting if the authors provide their insight on how LAP relates to canonical phagosome maturation in terms of effectivity, function, or other relevant aspects.
- The important role of PtdIns in phagosome formation and in early phagosomes is explained, however the roles of PtdIns that are formed subsequently in maturation towards late phagosomes and/or fusion with lysosomes is not mentioned. I believe adding this information would help explain the entire process from phagosome formation to maturation.
- Line 186: ‘Guanine nucleotide exchange factor (GEF)’, not quinine exchange factor
- Line 212: should read ‘Rho-associated kinase (ROCK)’, instead of Rho Kinase (ROCK)
- Line 315: “First, Mon1 is replaced with Rabex-5 in the phagosomal membrane and terminates the positive feedback loop of Rab5 activation”. In this sentence, ‘is replaced’ should read ‘replaces’
- Line 315: The reference to figure 5 is incomplete: ‘(Fig.’
Author Response
Comments:
The manuscript describes 1) the initiation of phagocytosis, including receptors and signalling pathways involved; 2) the maturation of phagosomes, including its molecular mechanisms; and 3) examples of how bacterial pathogens prevent the initiation of phagocytosis or maturation of phagosomes. This is a fine organisation of the manuscript. However, the current title (“Phagosome maturation: A key mechanism in innate immunity against intracellular bacterial infection”) and abstract are primarily focussed on the maturation aspect. I believe a title and abstract that capture the entire scope of the review would be more in place.
Ans) We greatly appreciate the reviewer’s comment. Based on the reviewer’s excellent suggestion, we have revised the title to ‘Formation and maturation of the phagosome: A key mechanism in innate immunity against intracellular bacterial infection’ and have made changes to the abstract in the revised manuscript.
(Page 1: line 2 and lines 20-25)
LC3-associated phagocytosis is a relatively novel process (first described by Martinez et al. in 2011, PMID 21969579) that was found to enhance the maturation process of phagosomes. I would expect this topic to be included in a review about phagosome maturation. Furthermore, it would be highly interesting if the authors provide their insight on how LAP relates to canonical phagosome maturation in terms of effectivity, function, or other relevant aspects.
Ans) In accordance with the reviewer’s comment, we have included LC3-associated phagocytosis (LAP), which is another type of phagocytosis process. Because LAP differs from canonical phagocytosis at an early stage, we have explained the difference between LAP and canonical phagocytosis during early phagosome formation.
(Page 9: lines 316-322; page 10: lines 323-334)
The important role of PtdIns in phagosome formation and in early phagosomes is explained, however the roles of PtdIns that are formed subsequently in maturation towards late phagosomes and/or fusion with lysosomes is not mentioned. I believe adding this information would help explain the entire process from phagosome formation to maturation.
Ans) We are thankful for the reviewer’s comment. PtdIns(3)P and PtdIns(4)P have been suggested as PtdIns that play an important role in phagosome maturation. Therefore, we have provided an additional explanation for the turnover of PtdIns during phagosome maturation.
(Page 10: lines 336-339, page 11: lines 389-394)
Line 186: ‘Guanine nucleotide exchange factor (GEF)’, not quinine exchange factor
The manuscript has been revised accordingly. (Page 5: line 194)
Line 212: should read ‘Rho-associated kinase (ROCK)’, instead of Rho Kinase (ROCK)
The manuscript has been revised accordingly. (Page 6: lines 221-222)
Line 315: “First, Mon1 is replaced with Rabex-5 in the phagosomal membrane and terminates the positive feedback loop of Rab5 activation”. In this sentence, ‘is replaced’ should read ‘replaces’
The manuscript has been revised accordingly. (Page 10: line 348)
Line 315: The reference to figure 5 is incomplete: ‘(Fig.’
The manuscript has been revised accordingly. (Page 10: line 349)
Comments:
The manuscript describes 1) the initiation of phagocytosis, including receptors and signalling pathways involved; 2) the maturation of phagosomes, including its molecular mechanisms; and 3) examples of how bacterial pathogens prevent the initiation of phagocytosis or maturation of phagosomes. This is a fine organisation of the manuscript. However, the current title (“Phagosome maturation: A key mechanism in innate immunity against intracellular bacterial infection”) and abstract are primarily focussed on the maturation aspect. I believe a title and abstract that capture the entire scope of the review would be more in place.
Ans) We greatly appreciate the reviewer’s comment. Based on the reviewer’s excellent suggestion, we have revised the title to ‘Formation and maturation of the phagosome: A key mechanism in innate immunity against intracellular bacterial infection’ and have made changes to the abstract in the revised manuscript.
(Page 1: line 2 and lines 20-25)
LC3-associated phagocytosis is a relatively novel process (first described by Martinez et al. in 2011, PMID 21969579) that was found to enhance the maturation process of phagosomes. I would expect this topic to be included in a review about phagosome maturation. Furthermore, it would be highly interesting if the authors provide their insight on how LAP relates to canonical phagosome maturation in terms of effectivity, function, or other relevant aspects.
Ans) In accordance with the reviewer’s comment, we have included LC3-associated phagocytosis (LAP), which is another type of phagocytosis process. Because LAP differs from canonical phagocytosis at an early stage, we have explained the difference between LAP and canonical phagocytosis during early phagosome formation.
(Page 9: lines 316-322; page 10: lines 323-334)
The important role of PtdIns in phagosome formation and in early phagosomes is explained, however the roles of PtdIns that are formed subsequently in maturation towards late phagosomes and/or fusion with lysosomes is not mentioned. I believe adding this information would help explain the entire process from phagosome formation to maturation.
Ans) We are thankful for the reviewer’s comment. PtdIns(3)P and PtdIns(4)P have been suggested as PtdIns that play an important role in phagosome maturation. Therefore, we have provided an additional explanation for the turnover of PtdIns during phagosome maturation.
(Page 10: lines 336-339, page 11: lines 389-394)
Line 186: ‘Guanine nucleotide exchange factor (GEF)’, not quinine exchange factor
The manuscript has been revised accordingly. (Page 5: line 194)
Line 212: should read ‘Rho-associated kinase (ROCK)’, instead of Rho Kinase (ROCK)
The manuscript has been revised accordingly. (Page 6: lines 221-222)
Line 315: “First, Mon1 is replaced with Rabex-5 in the phagosomal membrane and terminates the positive feedback loop of Rab5 activation”. In this sentence, ‘is replaced’ should read ‘replaces’
The manuscript has been revised accordingly. (Page 10: line 348)
Line 315: The reference to figure 5 is incomplete: ‘(Fig.’
The manuscript has been revised accordingly. (Page 10: line 349)
Reviewer 2 Report
This is an interesting review that I enjoyed reading.
The writing style is a bit repetitive and reads like a bit of a list so perhaps that should be addressed to keep the reader engaged.
I am not sure if the title is the best as it seems to highlight phagosome maturation. As initial steps of phagocytosis are also detailed, perhaps a more general title would be more appropriate?
I enjoyed the section on pathogens. If the authors wanted to expand on that, it could be interesting to discuss some pathogens that 'mimic' the internalisation pathways discussed to promote their uptake into non-phagocytic cells? Some description of the actions of other pathogens (Salmonella, Legionella) on phagosome maturation would be interesting and would complement the interesting Mtb discussion
As a minor point on line 186: quinine should probably read guanine
Author Response
Response to the Reviewer 2
Comments and Suggestions for Authors
This is an interesting review that I enjoyed reading.
The writing style is a bit repetitive and reads like a bit of a list so perhaps that should be addressed to keep the reader engaged.
I am not sure if the title is the best as it seems to highlight phagosome maturation. As initial steps of phagocytosis are also detailed, perhaps a more general title would be more appropriate?
Ans) We greatly appreciate the reviewer’s comment and have changed the title to ‘Formation and maturation of the phagosome: A key mechanism in innate immunity against intracellular bacterial infection’.
(Page 1: line 1)
I enjoyed the section on pathogens. If the authors wanted to expand on that, it could be interesting to discuss some pathogens that 'mimic' the internalisation pathways discussed to promote their uptake into non-phagocytic cells? Some description of the actions of other pathogens (Salmonella, Legionella) on phagosome maturation would be interesting and would complement the interesting Mtb discussion
Ans) We thank the reviewer for his/her interest in our paper. We have supplemented the description of the regulation of phagosome maturation and phagosomal acidification by several pathogens, including Legionella and Salmonella. (Page 13: lines 464-465 and lines 471-476; Page 14: lines 492-494 and lines 499-503)
Pathogens utilize various mechanisms to regulate and avoid the immune system in phagocytic as well as nonphagocytic cells. Based on the reviewer’s comment, the addition of expanded findings of immune evasion strategies or the regulation of immune systems in nonphagocytic cells could improve the quality of our paper. However, this review focuses on the regulation of phagocytosis in phagocytic cells during intracellular bacterial infection. Therefore, we would like to explain the mechanisms used to regulate the immune system in nonphagocytic cells, including mimicking the internalization pathway or changing the environment to favor pathogen survival, in a future review if we get the opportunity.
As a minor point on line 186: quinine should probably read guanine
The manuscript has been revised accordingly. (Page 5: line 194)